# Understanding MAPK Signaling Pathways in Apoptosis

**DOI:** 10.3390/ijms21072346

**Published:** 2020-03-28

**Authors:** Jicheng Yue, José M. López

**Affiliations:** 1School of Biology and Basic Medical Sciences, Soochow University Medical College, 199 Ren’ai Road, Suzhou 215123, China; jicheng_yue@suda.edu.cn; 2Institut de Neurociències, Departament de Bioquímica i Biologia Molecular, Unitat de Bioquímica, Facultad de Medicina, Universitat Autònoma de Barcelona, 08193 Cerdanyola del Vallès, Barcelona, Spain

**Keywords:** apoptosis, protein kinases, caspases, signaling, MAPK, p38, JNK, ERK, oocytes, feedback loops

## Abstract

MAPK (mitogen-activated protein kinase) signaling pathways regulate a variety of biological processes through multiple cellular mechanisms. In most of these processes, such as apoptosis, MAPKs have a dual role since they can act as activators or inhibitors, depending on the cell type and the stimulus. In this review, we present the main pro- and anti-apoptotic mechanisms regulated by MAPKs, as well as the crosstalk observed between some MAPKs. We also describe the basic signaling properties of MAPKs (ultrasensitivity, hysteresis, digital response), and the presence of different positive feedback loops in apoptosis. We provide a simple guide to predict MAPKs’ behavior, based on the intensity and duration of the stimulus. Finally, we consider the role of MAPKs in osmostress-induced apoptosis by using *Xenopus* oocytes as a cell model. As we will see, apoptosis is plagued with multiple positive feedback loops. We hope this review will help to understand how MAPK signaling pathways engage irreversible cellular decisions.

## 1. A Simple Introduction to MAPK Cascades

Over three decades have passed since Sturgill and Ray detected and characterized a new serine/threonine kinase, which was termed MAP kinase (MAPK) as it catalyzes the phosphorylation of microtubule-associated protein 2 (MAP-2) in insulin-treated 3T3-L1 adipocytes [1,2]. The MAP kinase was cloned two years later and named extracellular signal-regulated kinase 1 (ERK1) [3]. From then on, a super kinase family represented by ERK1 aroused a lot of interest in the scientific circles. As most family members are involved in processing signals stimulated by growth factors, the super kinase family was termed “mitogen-activated” protein kinase (MAPK) [4]. Members of the MAPK family share related structures and biochemical properties, especially as they are activated by dual phosphorylation on a tripeptide motif (Thr-X-Tyr) located in the kinase activation loop (T-loop) through a so called three-tiered MAPK cascade that comprises a MAPK, a MAPK kinase (MAPKK or MAP2K), and a MAPK kinase kinase (MAPKKK or MAP3K) [5]. On the contrary, MAPK inactivation is mediated by phosphatases through the dephosphorylation of threonine and/or tyrosine residues within the activation loop. Dephosphorylation can be achieved by serine/threonine phosphatases [6,7], tyrosine phosphatases (reviewed in [8]), or dual-specificity phosphatases (DUSPs) that dephosphorylate both the Thr and Tyr residues within the activation loop (reviewed in [9]). In addition, for accurate signal transmission both MAPK activation and inactivation are monitored by some specific scaffold proteins. In mammalians, 14 MAPK members divided into 7 subgroups have been identified, including four conventional MAPK subgroups that work in a typical three-tiered module, such as extracellular-regulated kinase (ERK1/2), C-Jun N-terminal kinase (JNK), p38 MAPK, and ERK5, and three atypical MAPK subgroups that do not follow the classical three-tiered, dual-phosphorylation signaling structure, such as ERK3/4, ERK7/8, and nemo-like kinase (NLK) (reviewed in [10]). 

Over the past 30 years, extensive research has shown that MAPKs play a pivotal role in converting extracellular stimuli into a wide range of cellular responses, including cell growth, migration, proliferation, differentiation, and apoptosis. Among them, JNK and p38 MAPK are activated most notably following cell exposure to stress evoked by a variety of physical, chemical, and biological stress stimuli, whereas ERK1/2 cascades mostly process cell growth factor-stimulated signaling [5,11]. Here, we present the recent progress about the role of the three prototypical MAPK groups in apoptosis. The dual nature of MAPKs in cell death (pro- or anti-apoptotic) and their dynamics will be discussed. We will consider the signaling properties of MAPKs in response to different stimuli, and the presence of multiple positive feedback loops that promote irreversible processes in the cell. Finally, we will explain the new insights into the MAPK function in apoptosis that were obtained by using *Xenopus* oocytes as a cell model. We hope this review will help to better understand the complex world of MAPKs.

## 2. Both JNK and p38 MAPK Cascades Mediate Pro-Apoptotic Processes

The stress-activated JNKs and p38 MAPKs play key roles in balancing cell survival and death in response to both extracellular and intracellular stresses (reviewed in [12]). Extensive research in apoptosis suggests that these kinases function in a cell context-specific and cell type-specific manner to integrate signals at different transmission points though both transcription-dependent and transcription-independent mechanisms, which would eventually converge on caspase activation. Generally, caspases can be activated by either an extrinsic pathway or an intrinsic pathway; the former is initiated by cell-surface death receptors stimulated by their corresponding ligands, and the latter is induced by protein release from the mitochondrial outer membrane due to pro-apoptotic Bcl-2 (B-cell lymphoma 2) family proteins mediating mitochondrial permeabilization [13]. Cytochrome c release from the outer mitochondrial membrane is a critical step in the intrinsic apoptotic pathway. Several Bcl-2 family proteins, both pro- and anti-apoptotic groups, are under the control of JNK and/or p38 MAPK cascades at a transcriptional and/or post-transcriptional level. Three different *jnk* genes and four different *p38* genes have been described in vertebrates, and more “flavors” can be obtained by selective transcription and alternative splicing. However, the role of different JNK and p38 isoforms in the regulation of apoptosis is not so clear and will not be treated in this review. We will consider only the role of JNK1-1 and JNK1-2 in osmostress-induced apoptosis (Section 8). 

### 2.1. Transcriptional Regulation

A diverse set of JNK and p38 MAPK substrates that promote apoptosis have been identified and validated [14,15]. A variety of transcription factors have been reported to be regulated by JNK and p38, which result in increased expression of pro-apoptotic proteins and decreased expression of anti-apoptotic proteins [14,16]. A major JNK target is transcription factor AP-1 (activator protein 1), a dimeric (homo- or heterodimer) complex that comprises members of the Jun (c-Jun, JunB, and JunD), Fos (c-Fos, FosB, Fra1, and Fra2), ATF (activating transcription factor), and MAF (V-maf musculoaponeurotic fibrosarcoma) protein families. The diverse combinations of AP-1 determine distinct gene transcriptional profiles under the control of JNK and/or p38 MAPK cascades. For example, c-Jun can be phosphorylated by JNK [17] and p38 [18], and the activated c-Jun can autoregulate its own expression in a positive regulatory loop through a c-Jun/AP-1 enhancer element in its promoter [19]. AP-1 regulates a wide range of cellular processes, including cell proliferation, differentiation, cell survival, and apoptosis [20,21]. Although AP-1 activation is associated with apoptotic scenarios, its role in ensuring cell survival seems equally important. The pro- or anti-apoptotic role of AP-1 activation seems to be dependent on the cellular and extracellular context [22]. In addition to AP-1, one of the best-known transcription factors regulated by JNK and p38 MAPK cascades in apoptosis is p53 tumor suppressor protein. In stressed cells, JNK-mediated phosphorylation can stabilize and activate p53 and thus promote programmed cell death [23]. Like the c-Jun component of AP-1, the transcription factor p53 works in combination with other proteins. It was reported that p53-p73 dimerization is critical in the induction of apoptotic cell death, particularly in response to the JNK-mediated cell stress response. Activated JNK phosphorylates p53 at Thr81 in the proline-rich domain (PRD), which enables the dimerization of p53 and p73. The p53-p73 dimer facilitates the expression of several pro-apoptotic target genes, such as *puma* and *bax* [24]. However, in HIV-1 envelope glycoprotein complex (Env)-induced apoptosis, p38 MAPK promotes cell death through phosphorylation of p53 at Ser46 instead of Thr81 [25], which may imply a distinct dimerization status of p53. An identified p53 partner protein targeted by p38 MAPK cascade is p18 Hamlet [26]. In response to genotoxic stresses induced by UV or cisplatin treatment, p18 Hamlet is phosphorylated and stabilized. Phosphorylated p18 Hamlet dimerizes with p53 and stimulates the transcription of several pro-apoptotic p53 target genes, such as *puma* and *noxa* [26]. It has also been reported that p53 is positively self-regulated in a JNK-dependent manner via suppression of Wip1 (wild-type p53-induced phosphatase 1), a p53 inhibitor encoded by the gene *ppm1d* [27]. Even if p53 is a major JNK/p38 MAPK substrate in promoting apoptosis, in some contexts, p53 is not phosphorylated by activated JNK or p38 MAPK. In eIF5A1 overexpression-induced cell death, JNK and p38 MAPK cascades promote apoptosis independently of p53 activation [28]. Similarly, in LPS/D-Gal (lipopolysaccharide/d-galactosamine)-induced liver injury, the AMPK (5′ adenosine monophosphate-activated protein kinase) signaling pathway promotes apoptosis through JNK activation, but the level of p53 remains unchanged [29]. Besides AP-1 and p53, other transcription factors are implicated in MAPK-induced apoptosis. For instance, JNK-catalyzed phosphorylation can promote the release of the transcription factor FoxO1 (forkhead box protein O1) from its blocking protein 14-3-3 in oxidative stress-induced mouse follicular granulosa cell apoptosis [30]. These data support the idea that MAPK signaling pathways function in a cell context-specific and cell type-specific manner to promote apoptosis. 

### 2.2. Post-Transcriptional Modifications

In addition to regulating the levels of the expression of apoptotic proteins, JNK/p38 MAPK cascades also regulate their functions directly in both extrinsic and intrinsic apoptotic pathways. The extrinsic pathway, or death-receptor pathway, can induce caspase-8 activation and apoptosis independently of Bcl-2 family members. However, in some cells, the extrinsic pathway can induce caspase-8-mediated cleavage of the pro-apoptotic BH3-only protein Bid (BH3 interacting-domain death agonist). The C-terminal truncated form of Bid (tBid) translocates to mitochondria and induces the release of cytochrome c, which in turn induces further caspase activation (caspase-9 and the effector caspases-3, -6, and -7) through the intrinsic pathway [31]. Therefore, the extrinsic and intrinsic pathways of apoptosis can converge in the mitochondria. It has been reported that JNK phosphorylates the E3 ubiquitin ligase iTCH (itchy homolog), which in turn promotes ubiquitination-mediated degradation of the caspase-8 inhibitor c-FLIP (cellular FLICE (FADD-like IL-1β converting enzyme) inhibitory protein) [32]. Intriguingly, TNFα (tumor necrosis factor alfa) treatment activates the JNK cascade, which induces caspase-8-independent cleavage of Bid; the resultant fragment of Bid (jBid), distinct to t-Bid, translocates to mitochondria and selectively promotes the release of Smac/DIABLO (second mitochondria-derived activator of caspases/direct inhibitor of apoptosis protein-binding protein with low pI) [33]. It has also been reported that JNK protects Bid from caspase-8-mediated cleavage through the phosphorylation of Bid at Thr59, but full-length phosphorylated Bid is cleaved by an unknown protease to generate jBid [34]. More studies are necessary to identify this protease and uncover its function during cell death.

p38 and JNK can also regulate autophagy programs. The activation of RIPK (receptor-interacting serine/threonine-protein kinase) and JNK can induce autophagic cell death [35]. Although autophagy was initially identified as a cell survival mechanism during nutrient starvation, autophagy plays high context-specific roles in mediating cell death. The interplay between autophagy-dependent cell death and other types of cell death was reviewed recently [36,37]. The role of JNK in autophagy and other types of cell death was also reviewed [38]. Recently, it has been reported that sustained p38α activation induces autophagy but determines that cancer cells preferentially enter senescence instead of apoptosis [39].

In the intrinsic pathway, JNK-mediated phosphorylation of 14-3-3 protein, a cytoplasmic anchor of several proteins, induces Bax (Bcl-2-associated X protein) release and translocation to the mitochondria [40], and the release of the pro-apoptotic proteins Bad (Bcl-2-antagonist of cell death) and FOXO3a [41]. JNK and p38 can directly phosphorylate and regulate the function of several members of the Bcl-2 protein family. JNK can induce the phosphorylation of Bad at Ser128 to promote the apoptotic effect of Bad [42]. p38 and JNK induce Bax phosphorylation and mitochondrial translocation to promote apoptosis [43]. The pro-apoptotic activity of Bim (Bcl-2-like protein 11) depends on the phosphorylation at Ser65 catalyzed by JNK [44] or p38 MAPK [45]. During UV-induced apoptosis, Bim and Bmf (Bcl-2-modifying factor), normally sequestered by dynein and myosin V motor complexes, are released by JNK-dependent phosphorylation [46]. Bim and Bmf are BH3-only proteins that interact with Bcl-2-like molecules to neutralize their anti-apoptotic effect.

Additionally, JNK and p38 may inhibit the functions of the anti-apoptotic Bcl-2 family members. It has been shown that JNK can phosphorylate Bcl-2 [47,48] and Mcl-1 (myeloid cell leukemia 1) [49], thereby suppressing their anti-apoptotic function. In Fas-induced cell death, p38 activation attenuates the accumulation of the anti-apoptotic proteins Bcl-xL (B-cell lymphoma-extra large) and Bcl-2 in the mitochondria [50]. Figure 1 summarizes all the known JNK substrates that promote cytochrome c release from the mitochondria.

## 3. JNK and p38 MAPK Cascades Also Mediate Anti-Apoptotic Processes

JNK and p38 MAPK cascades not only promote pro-apoptotic processes; they are involved in anti-apoptotic mechanisms as well. There are many examples of anti-apoptotic processes mediated by transcriptional regulation. BNIP3 upregulation, through activation of the ERK and JNK pathways, is required for the protection of keratinocytes from UVB-induced apoptosis [51]. JNK activation attenuates endoplasmic reticulum (ER)-induced cell death due to enhanced expression of several anti-apoptotic proteins [52]. ER stress induced by thapsigargin and tunicamycin leads to increased expression of TRPV6 (transient receptor potential vanilloid channel 6) via the JNK signaling pathway, which could protect human embryonic stem cell-derived cardiomyocytes (hESC-CMs) from apoptotic cell death [53]. Besides, the JNK/c-Jun signaling pathway promotes annexin A2 overexpression that would suppress the expression of p53, and thus would decrease p53-regulated apoptotic genes [54]. In oxidized low-density lipoprotein (oxLDL)-induced macrophage apoptosis, p38 MAPK signaling pathway activation induces orphan nuclear receptor Nur77 expression that enhances cell survival via the suppression of apoptosis [55]. 

At post-translational levels, it was well established that phosphorylation of caspase-9 at Thr125 by ERK1/2 restrains intrinsic apoptosis in mitosis [56]. However, during the acute response to hyperosmotic stress in mouse embryonic fibroblasts, the inhibitory phosphorylation is catalyzed by p38 MAPK instead of ERK1/2 [57]. It was also reported that transient activation of JNK delays caspase-9 activation by direct interaction of JNK with Apaf 1 and cytochrome c to inhibit the apoptosome complex [58]. Another example of JNK’s anti-apoptotic function is the phosphorylation of Bad at Thr201 that promotes dissociation of the anti-apoptotic protein Bcl-xL from Bad [59]. It was reported recently that p38 MAPK/MK2 interacts with RIPK1 in a cytoplasmic complex and MK2 phosphorylates RIPK1 at Ser321/336 in response to pro-inflammatory stimuli and pathogens, which results in the inhibition of RIPK1 auto-phosphorylation and thus curtails RIPK1-dependent apoptosis and necroptosis [60]. 

Therefore, as we have seen in the previous sections, the activation of JNK and p38 signaling pathways can induce or prevent apoptosis. In Section 6, we will present an easy guide to understand the MAPK signaling function.

## 4. MAPK Signal Cross-Talks: JNK and p38 Do Not Always Work in Tune

It could seem that JNK and p38 always work in tune, but things are more complicated. Crosstalk signaling between JNK and p38 MAPK is emerging as an important regulatory mechanism in stress responses. The JNK and p38 MAPK pathways share several upstream regulators, and accordingly there are multiple stimuli that simultaneously activate both pathways; however, there is evidence indicating that the p38 MAPK pathway can negatively regulate JNK activity in several contexts. Chemical inhibition of p38 strongly increased the activation of JNK [61], and deletion of the *p38α* gene induced JNK activation in a mouse model [62]. Wada et al. reported that the balance between JNK and p38 signaling pathways determines cell fate and links environmental and developmental stress to cell cycle arrest, senescence, oncogenic transformation, and adult tissue regeneration [63]. These authors also showed that antagonistic regulation of Cdc2 by JNK and p38 may be one of the checkpoints by which different stress-signaling pathways control the cell fate. Staples et al. demonstrated that UV-C-induced p38 activation increases the levels of MAPK phosphatase-1 (DUSP1) in mouse embryonic fibroblasts (MEFs), and DUSP1 protects these cells from UV-C-induced apoptosis by inactivating JNK [64]. More recently, by analyzing the single cell response to UV stress, it has been reported that p38-induced DUSP1 is inversely correlated with JNK activity, and this cross-inhibition generates cellular heterogeneity in JNK activity [65]. Although p38 causes variation of the DUSP1 protein levels between cells, the molecular mechanism is not so clear (see the concluding remarks). Anyway, cells are sentenced to death when JNK activity surpasses a threshold level [65].

## 5. The Two Faces of ERK in Apoptosis

ERK1/2 activation is widely associated with anti-apoptotic functions by controlling cell proliferation and differentiation [11,66]. ERK can be anti-apoptotic by the downregulation of pro-apoptotic proteins and upregulation of anti-apoptotic proteins through both transcriptional and post-translational mechanisms (reviewed in [67]). However, there are several examples showing that ERK1/2 signaling can be pro-apoptotic (reviewed in [68]). 

Both cell cycle arrest and cell cycle re-enter can induce apoptosis. Aspafilioside B induces G2 phase cell cycle arrest and apoptosis through ROS-independent ERK and p38 MAPK activation [69]. It has been reported that both the G2 phase delay and blocked exit from the G2 checkpoint arrest are mediated by the MEK1-dependent destabilization of the critical G2/M regulator Cdc25B [70]. Postmitotic neurons, arrested in the G0 state, make attempts to re-enter the cell cycle in response to stress signals, which results in cell death. For example, it has been reported that β-amyloid peptide induces increased expression of cyclin D1 mediated by MEK-ERK activation, therefore promoting S-phase entry and neuronal cell death [71]. 

Different insults can induce apoptosis through activation of the ERK signaling pathway. DNA damage induced by cisplatin activates ERK1/2 and increases p53 protein levels [72]. The authors also found that ERK1/2 interacts with p53 and induces the phosphorylation of p53 at Ser15. Therefore, upregulation of p53 by the ERK signaling pathway may be one of the mechanisms of DNA damage-induced apoptosis. Activation of ERK1/2, JNK, and p38 MAPK with different kinetics were detected in perfluorohexanesulfonate (PFHxS)-induced neurotoxicity of PC12 cells. The ERK inhibitor significantly reduced apoptosis, while JNK inhibition significantly increased apoptosis, whereas the p38 MAPK inhibitor had no effect [73]. ERK1/2 activation also contributes to the anti-proliferation and apoptotic effects of NSC 95397 (a quinone-based small molecule compound) in colon cancer cells [74]. Glutamate-induced oxidative toxicity in neurons is associated with persistent ERK activation, and ERK inhibition protects cells from glutamate toxicity [75]. Combined TNFα and NMDA (N-methyl-d-aspartate) stimulation induces ERK signaling and cell death in mouse cortical neurons, which can be protected with MEK/ERK inhibition [76]. However, care should be taken when the conclusions are based, exclusively, on the effects of chemical inhibitors. Recently, it has been reported that ERK can facilitate apoptosis in response to oxidative stress and oxygen-glucose deprivation through phosphorylation of the pro-fusion protein mitofusin (MFN) 1, as phosphorylated MFN1 has a high affinity to the pro-apoptotic Bcl-2 family member Bak (Bcl-2 homologous antagonist killer) [77]. 

## 6. An Easy Guide to Understand MAPKs-Regulated Apoptosis

Taking into account the diverse and contradictory effects of MAPKs on the regulation of apoptosis, can we present some simple rules to understand the behavior of MAPKs? Is it possible to predict whether a stress response that activates MAPKs would end up in cell death? In other words, can we say something insightful without recurring to the common mantra “MAPK function in apoptosis depends on the cell type and the stimuli”? Let’s try it.

First, MAPKs must have signaling properties well suited for the processing and propagation of external or internal stimuli that, in the face of a noxious situation, will determine whether a cell will survive and repair the damage or will die by apoptosis. The kinetics and the intensity of the signaling pathway activated is going to be important, as well as the presence of signaling loops. Next, we are going to describe the basic properties of MAPKs that are decisive on whether to engage a cell death program, and the positive feedback loops implicated. 

### 6.1. Basic Signaling Properties of MAPKs

Functioning as cellular sensors and switches, MAPKs play a critical role in the regulation of cell fate decisions. Cellular sensors must have signaling properties well suited for the processing and propagation of stimuli that promote irreversible processes. Basically, these properties are: (1) Ultrasensitivity, meaning that a small increase in stimulus produces a very large response after a threshold is crossed; (2) hysteresis, a form of biochemical memory reflected in the sustained activation of MAPK when the stimulus has disappeared; and (3) digital response, meaning an all-or-none response at a single cell level [78] (Figure 2). 

Ultrasensitive responses are commonplace in many biological processes. At least four different mechanisms can generate ultrasensitive responses. These include zero-order ultrasensitivity, multisite ultrasensitivity, inhibitor ultrasensitivity, and positive feedback loops [79,80]. Positive and negative feedback loops are also common regulatory elements in biological signaling systems. Positive feedback is defined as a set of regulatory steps that feeds the output signal back to the input [81] (Figure 2D). Ultrasensitive systems embedded in a positive feedback loop have the potential to exhibit bistable behavior, switching between discrete stable steady states without being able to rest in intermediate states [82,83,84]. The three hallmarks of a bistable system are strong ultrasensitivity, digital response at the individual cell level, and hysteresis. Examples of such systems are the response of JNK to hyperosmotic shock, and the response of ERK to progesterone, which were described by the elegant works of James E. Ferrell in *Xenopus* oocytes, a suitable cell model to study these properties [85,86]. We also reported the basic properties of the p38 signaling pathway in response to hyperosmotic shock by using *Xenopus* oocytes. The response of p38 to osmostress was ultrasensitive, bimodal, and with low hysteresis [87]. 

In mammalian cells, JNK signaling also showed ultrasensitivity in response to different stimuli, and a switch-like response at a single cell level but no hysteresis [88]. It is also interesting to note that in mammalian cells, ERK activation by PDGF (platelet-derived growth factor) was not ultrasensitive, and showed a graded response at a single cell level, but the graded activation of ERK was converted into a more switch-like behavior (digital response) downstream of the kinase cascade, at the level of immediate–early gene induction [89]. The authors also found that, whereas ERK activation was graded, ERK localization into the nucleus presented an all-or-none (digital) behavior, suggesting that an unknown mechanism that controls nuclear translocation and/or retention of ERK is likely responsible for the ultrasensitive responses downstream. In *Saccharomyces*, it has been described that Hog1 (high osmolarity glycerol 1), the homologous of p38, gradually accumulates in the nucleus after increasing the salt concentration, but the transcriptional output obtained is bimodal [90].

Since switch-like responses obtained at a single cell level are common in MAPKs and other protein kinases, we proposed a digital model for cell decisions. Using the analogy of cells as information devices and of protein kinases as transistors or chips in computers, we can imagine networks of kinases that generate digital information to “run” different programs (like the cell death program). Therefore, a general model for sensing, integrating, and making choices, based in the generation of digital responses, may be applied to different biological processes [78].

### 6.2. Strong Versus Weak and Sustained Versus Transient Signaling

The ultrasensitive properties of MAPKs and the existence of threshold levels for the activation of downstream targets can explain the digital outputs obtained after MAPK activation. We observed that the response of JNK to hyperosmolar sorbitol is switch-like in character, and that an initial graded response is converted into digital after the critical period of cytochrome c release [91]. As we will see later, there are multiple feedback loops that might contribute to the digital response observed. Two important factors that regulate apoptosis are the strength and the duration of the signal. It is well known that the pro- or anti-apoptotic effects of MAPK activation depend on the duration of the signal. For instance, transient JNK activation is shown to promote cell survival, whereas prolonged JNK activation induces apoptosis [92]. The duration of phosphorylation of p38 also seems crucial to regulate cell fate, since sustained activation is associated with apoptosis [93], whereas transient phosphorylation is associated with survival [94]. For example, in SKT6 cells exposed to osmotic or heat shock, transient activation of JNK and p38 MAPK promotes erythroid differentiation, whereas prolonged activation of JNK and p38 MAPK causes apoptosis [95]. In yeast, sustained activation of p38 (Hog1) results in a severe growth defect and cell death by promoting high levels of reactive oxygen species (ROS) [96]. Persistent activation of ERK1/2 promotes glutamate-induced oxidative toxicity in neuronal cells [75]. It has been proposed that the main cause of sustained ERK activation is the presence of ROS during the cell death program [68]. However, sustained activation of ERK signaling is not only involved in apoptosis but also in cell cycle progression, cellular transformation, and differentiation [97]. 

Although it is generally accepted that strength and long signal duration are relevant to engaging apoptosis, this is not well explained at a molecular level. How are different targets that regulate cytochrome c release and caspase activation sensitive to the kinetics and strength of MAPK signaling? We propose that strong MAPK activation (mainly JNK and p38) might be enough to surpass a threshold level that would activate specific pro-apoptotic targets. Similarly, sustained activation of MAPKs could activate additional targets that are not usually activated by transient signaling. The affinity of the targets for activated MAPKs could be an important factor to determine which targets are activated first (anti-apoptotic) and which are activated later (pro-apoptotic). In other words, the pro-apoptotic targets would have low affinity for MAPKs, but they could be activated when MAPK activity is high and/or sustained, meaning that an increased amount of activated MAPK can interact with additional targets. 

For example, it has been reported that an increasing concentration of constitutively active MKK6 differentially activates either p38α alone (low MKK6 activity) or both p38α and p38γ (high MKK6 activity), both in vitro and in injected oocytes [98]. Interestingly, it has been shown that the temporal ordering of cell cycle phosphorylation depends on intrinsic properties of the substrate proteins: Good kinase substrates tend to be phosphorylated early, and good phosphatase substrates tend to be phosphorylated late [99,100,101]. 

### 6.3. Feedback Loops Could Explain Complex Biological Processes

Douglas R. Hofstadter, in his insightful book “*I Am a Strange Loop*” published in 2007 [102], proposed that consciousness could arise in the brain as a consequence of intricate interactions between symbols, which are not yet well defined at a molecular level but could be represented as clusters of activated neurons. Hofstadter also proposed that the presence of multiple feedback loops in this complex interactive network of symbols might be responsible for the sense of “I” or consciousness in evolved brains.

At first sight, feedback loops could seem strange, or even dangerous, for most people as Hofstadter points out, but scientists are more familiar with them. Feedback control is widespread in different biological systems and is very important to regulate intercellular signaling during development (reviewed in [103]). Feedback loops are also common in intracellular signaling to modulate cell responses in space and time and to control cell fate [81]. Negative feedback loops are well known in cellular systems and are necessary to maintain homeostasis. The first demonstration of an end-product feedback inhibition of a biosynthetic pathway (cholesterol homeostasis) was due to Rudolph Shoenheimer in 1933 [104]; a more recent example applies to the negative feedback loops that control iron homeostasis [105]. Negative feedback loops can also produce oscillations [84,106]. Indeed, many examples of positive feedback loops have been reported, and these loops can amplify a signal, change the timing of a signaling response, and create bistable switches [84,106,107]. As we will see next, these positive feedback loops are fundamental to explaining the irreversibility of the apoptotic process. 

### 6.4. How Cells Die through Positive Feedback Loops

The ultrasensitive response and the sustained activation of MAPKs, when the stimulus has disappeared, could be a consequence of the engagement of positive feedback loops when a threshold is reached. During the cell death program, multiple positive feedback loops have been reported (Figure 3). Cytochrome c release from the mitochondria induces caspase-9 and subsequent caspase-3 activation, which in turn produces Bcl-2 proteolysis, thus promoting more cytochrome c release and caspase-9/-3 activation in a positive feedback loop [108]. Bid proteolysis by caspase-8 produces tBid, which induces cytochrome c release, caspase-9, and caspase-6 activation, which in turn promotes proteolysis and activation of caspase-8, closing the vicious loop [109]. A similar loop has been reported by proteolysis of Bid induced by caspase-3 [110,111]. Caspase-3 also induces direct proteolysis and full activation of caspase-9 and was considered a positive feedback loop [112]. However, Denault et al. found that dimerization of caspase-9 within the apoptosome complex (leading to its autocatalytic cleavage) is required for subsequent cleavage by caspase-3, thus releasing the inhibitory effect of XIAP (X-linked inhibitor of apoptosis protein) on caspase-9 [113]. These authors refer to this mechanism as derepression instead of feedback activation [113], although this is a moot point [114]. Pro-caspase-3 is also a substrate of caspase-8 [115], making the caspase activation network more complex (Figure 3A). Recently, a systematic analysis of the individual and combined roles of caspases in chemotherapy-induced cell death within human leukemic cell lines, using CRISPR-Cas9 gene targeting, has shown that the activation of effector caspase-3 or -7 produces feedback amplification and efficient apoptotic cell death by inducing full activation of apical caspases (caspase-8 and -9) [116]. This study emphasizes the important role of effector caspases in engaging positive feedback loops for efficient cell death.

Very often, caspases are substrates of kinases and kinases are substrates of caspases in a death grip [117]. MEKK1 (MAPK/ERK kinase kinase 1), a MAPKKK that ultimately can activate JNK and p38 [118], is cleaved by caspase-3 during anoikis, Fas stimulation, and DNA damage [119,120,121]. The fragment obtained is a constitutively active MEKK1 that promotes cytochrome c release and caspase-9/-3 activation in turn (Figure 3A). 

In the JNK/p38 MAPK cascades, the second messenger reactive oxygen species (ROS) regulates MAPK activation in a positive circuit. It has been reported that ROS, generated by NADPH oxidase, activates JNK, which in turn activates NADPH oxidase in a positive feedback loop [122]. It has also been reported that the activation of NADPH oxidase 2 in cadmium-induced apoptosis activates JNK through ROS generation and protein phosphatase 5 (PP5) inactivation, and JNK activation in turn increases the expression of NADPH oxidase and its regulatory proteins (p22phox, p40phox, p47phox, p67phox, and Rac1), which promotes ROS generation, thus creating a positive feedback loop [123]. Shi et al. reported that ROS generation induces JNK activation, which in turn induces ROS generation through p53 activation [27]. A similar feedback loop between p53 activation, ROS generation, and p38 activation that in turn induces p53 activation was observed in cisplatin-induced apoptosis [124]. Sustained p38 activation induces important metabolic changes and enhances the respiration rate, thus increasing the production of mitochondrial ROS, which contributes to p38-induced apoptosis [125]. In isoliensinine-induced apoptosis of triple-negative breast cancer cells, the generation of ROS activates both the JNK and p38 MAPK signaling pathways, and p38 activation also induces ROS elevation in a positive feedback loop [126].

ASK1 (apoptosis signal-regulating kinase 1) is a MAP kinase kinase kinase (MAP3K) of JNK and p38 MAPK, which is preferentially activated by ROS (reviewed in [127]). Thus, *ask1*−/− embryonic fibroblasts are resistant to H_2_O_2_-induced apoptosis and sustained activation of JNK and p38 MAPK is lost [93]. Recently, it was reported that TNFR1 (tumor necrosis factor receptor 1) stimulated by ω-3-17,18-epoxyeicosanoic acid (C20E) activates the ASK1-MKK4/7-JNK/p38 MAPK signaling pathway, which results in Bid cleavage [128].

ASK1 is involved in different positive feedback loops. TNFα treatment induces the accumulation of Daxx (death-associated protein 6) protein by preventing its proteasome-dependent degradation in an ASK1-mediated pathway, and Daxx induces the activation of ASK1 [129]. In gastric cancer, ASK1 overexpression induces the transcription of cyclin D1 through AP-1 activation, and ASK1 levels are regulated by cyclin D1 in turn via the Rb-E2F (retinoblastoma-E2F transcription factor) pathway [130]. 

Besides ROS, ceramide is another second messenger associated with apoptosis. It has been reported that ceramide can initiate apoptosis through Rac1- or ASK1-regulated activation of the JNK/p38 MAPK cascades [131]. The JNK signaling pathway stimulates ceramide generation and promotes apoptosis by phosphorylation of neutral sphingomyelinase (nSMase) [132]. Therefore, a positive feedback loop exists between JNK activation and ceramide production. In summary, different positive feedback loops could promote sustained activation of the JNK/p38 signaling pathways (Figure 3B).

## 7. *Xenopus* Oocytes as a Cell Model to Understand Apoptosis 

*Xenopus* oocytes are a very useful system to study cell death mechanisms. The cell-free system based on *Xenopus* egg extracts was used in the pioneering experiments describing the mitochondrial apoptotic pathway and the role of cytochrome c release in caspase activation [133,134]. Moreover, *Xenopus* oocytes can be used as an in vivo system since these cells, when submitted to different stresses, initiate the mitochondrial pathway of apoptosis. Expression of Bcl-x(S) [135], oocyte nutrient depletion induced through inhibition of the pentose phosphate pathway [136], neutral sphingomyelinase-induced ceramide [137], or hyperosmotic shock [91] promotes apoptosis in *Xenopus* oocytes. These cells might be appropriate for the study of metabolic regulation of cancer since the oocytes exhibit altered metabolism coupled to its apoptotic machinery in a similar fashion to cancer cells [138]. Besides environmental stimuli, the apoptotic machinery can be turned on spontaneously during meiotic maturation in unfertilized *Xenopus* eggs [139].

*Xenopus* oocytes allow greater manipulation than other systems and are especially suitable for biochemical determinations at a single cell level. For instance, it is possible to measure by Western blot the activation of MAPKs and the release of cytochrome c in single cells. Capped RNAs are microinjected in the oocytes to express, with high efficiency, the constitutively active or dominant negative mutants of the protein kinases. It is also possible to microinject antibodies, antisense oligonucleotides, or other molecules that cannot cross the cell membrane in order to manipulate any specific signaling pathway. Importantly, cytochrome c microinjection in *Xenopus* oocytes induces caspase-3 activity [140], making affordable the study of the signaling pathways activated by cythochrome c release and the positive feedback loops engaged by caspases [87,111]. Recently, by using *Xenopus* oocytes and egg extracts, it was demonstrated that positive feedback loops allow apoptosis to spread though the cytoplasm in self-regenerating trigger waves, which were correlated with caspase activation [141].

## 8. MAPK Dynamics in Hyperosmotic Shock-Induced Apoptosis

Hyperosmolarity has many damaging effects on cells by promoting water flux out of the cell, triggering cell shrinkage and intracellular dehydration [142]. Cells have developed survival mechanisms to adapt osmotic changes [143], but when the stress is intense or persistent, it triggers apoptosis [144,145,146,147,148].

During the last years, we analyzed the mechanisms that regulate osmostress-induced apoptosis in *Xenopus* oocytes. We reported that hyperosmotic stress induces cytochrome c release and caspase-3 activation [91]. We have also described the time-course events during the apoptotic program and the role of stress protein kinases, calpains, and Smac/DIABLO. Hyperosmotic shock induces rapid calpain activation and high levels of Smac/DIABLO release from the mitochondria (early events) before significant amounts of cytochrome c are released to promote caspase-3 activation (late events) [149]. Hyperosmotic shock also activates the p38 and JNK signaling pathways very quickly [87,91]. Simultaneous inhibition of both p38 and JNK pathways reduces osmostress-induced apoptosis, while sustained activation of these kinases accelerates the release of cytochrome c and caspase-3 activation. Therefore, at least four different pathways induced early by osmostress converge on the mitochondria to trigger apoptosis [149].

Recently, we characterized the JNK isoforms activated by hyperosmotic shock and their role in apoptosis. Transcripts derived from the *jnk1* gene present alternative splicing at the C terminus, yielding short and long JNK1 variants (JNK1-1 and JNK1-2, respectively). We have shown that JNK1-1 and JNK1-2 are activated early by osmostress and that sustained activation of both isoforms accelerates the apoptotic program. Later, when caspase-3 is activated, JNK1-2 is proteolyzed at Asp385, increasing the release of cytochrome c and caspase-3 activity, and therefore creating a positive feedback loop [111]. Since cleaved JNK1-2 and JNK1-1 have similar amino acid sequences except for the last five at the *C*-terminal, and the expression of the short JNK1-2 fragment has a clear apoptotic effect compared with JNK1-1, it implies that the last amino acids of the JNK1-2 fragment might be important for regulating cytochrome c release and caspase-3 activation in the positive feedback loop engaged after JNK1-2 cleavage [111]. 

In addition, an unknown protease induces early proteolysis of Bid and mono-ubiquitinated Bid at Asp52, which in turn increases the release of cytochrome c and caspase-3 activity, which in turn produces massive proteolysis of Bid, thus creating a second positive feedback loop [111]. Interestingly, we have also reported that cytochrome c microinjection in *Xenopus* oocytes induces phosphorylation of p38 through caspase-3 activation, and caspase inhibition reduces the level of p38 activation induced by osmostress, suggesting that a third positive feedback loop is engaged by hyperosmotic shock [87]. In addition, it has been reported that caspases can induce proteolysis of the calpain inhibitor calpastatin [150,151], thus increasing calpain activation in a positive loop. In summary, hyperosmotic shock induces the activation of different pathways that converge on the activation of caspase-3, which engages an irreversible apoptotic program through the activation of multiple positive feedback loops. 

A model for osmostress-induced apoptosis in *Xenopus* oocytes is presented in Figure 4. The model is composed of two different phases: A reversible and an irreversible phase. In the reversible phase, anti-apoptotic and pro-apoptotic responses occur synchronically, including the rapid release of Smac/DIABLO, calpain activation, early Bid cleavage, and JNK/p38 MAPK phosphorylation. These pathways are going to converge on the mitochondria to trigger caspase-3 activation. Oocytes are not committed to death in the reversible phase and could recover when the stress is removed. MAPKs (JNK and p38) are sensors of stress that would activate early anti-apoptotic substrates (A, B, C), but strong and/or sustained activation of MAPKs would activate late pro-apoptotic substrates (X, Y, Z). These substrates, not yet characterized, could include different Bcl-2 family members. Mitochondria integrate the available information to regulate cytochrome c release, which, after reaching a threshold level in the cytosol, activates caspase-3 that engages multiple positive feedback loops to produce an irreversible death.

## 9. Concluding Remarks

MAPKs regulate apoptosis through transcriptional and post-transcriptional mechanisms. Although p38 and JNK activation are usually associated with pro-apoptotic effects and ERK activation with anti-apoptotic effects, examples of the opposite were also presented here. Moreover, MAPK signal crosstalk between JNK and p38 MAPK is an additional regulatory mechanism in stress responses. The dual role of MAPKs in cell death (pro- and anti-apoptotic) can be partially explained by the diversity of targets activated through different stimuli, or by the molecular context where a specific stimulus is acting (the cell type considered). However, even in a certain cell type exposed to a specific stimulus, the activation of a MAPK signaling pathway can be pro- and/or anti-apoptotic. This paradox can be explained, in part, by the basic properties of MAPK signaling systems. The ultrasensitive properties of MAPKs and the existence of threshold levels for activation explain why strong and/or sustained activation of MAPKs is pro-apoptotic, whereas weak and/or transient activation is anti-apoptotic. Further studies about the temporal ordering of phosphorylation, early for anti-apoptotic targets and late for pro-apoptotic targets, are necessary to better understand the behavior of MAPKs. It seems that the differential affinity of substrates for MAPKs could account for the kinetics and dual nature of MAPKs. A second paradox is why some cells submitted to the same dose of stimulus during the same time can survive, whereas others are committed to cell death. The signaling properties of MAPKs, including the presence of multiple positive feedback loops, explain why single cells exposed to intermediate levels of stress present all-or-none (digital) responses. However, the molecular reason for this heterogeneous response is not so clear. As we have pointed out before, it has been reported that UV stress generates cellular heterogeneity in JNK activity through variations in the protein levels of MAPK phosphatase-1 (DUSP1) via p38 activation [65]. However, it is unclear how the DUSP1 induction varies between cells. It has been proposed that the intrinsic noise of gene expression produces cellular heterogeneity that is known as non-genetic heterogeneity [152]. Interestingly, it has been shown that cells present a heterogeneous composition in pro- and anti-apoptotic factors, which are the primary causes of cell-to-cell variability in the timing and probability of death in TRAIL (TNF-related apoptosis-inducing ligand)-induced apoptosis [153]. It is likely that a multivariate control due to different proteins and signaling pathways, including MAPKs, is operating in most types of cell death. Therefore, the prediction of the fate of single cells exposed to stress might be difficult, but not impossible, since it would require integrating the information obtained from different factors.

Here, we described the presence of multiple positive feedback loops in apoptosis. These loops are fundamental to the creation of self-sustained trigger waves of activity (caspases and/or kinases) in space and time. Figure 5 is a simplified version of the complex loops of signaling previously described. Such a modular structure could be a general mechanism to attain irreversible changes in the cell, and therefore might be applied to other biological processes, although the kinases and proteins involved could be different. For instance, the ERK signaling pathway is activated by progesterone in *Xenopus* oocytes and regulates meiotic progression through activation of the Cdc2/Cyclin B (MPF) complex, although JNK is not involved [154]. Several positive feedback loops have been described for this biological process that assure sustained activation of ERK2 and MPF to promote an irreversible maturation of the oocyte [155,156,157,158].

A review on MAPKs will always show a puzzling image. This is not surprising at all, since MAPKs are involved in multiple processes and cellular signaling is a complex world. However, here, we have tried to extract and distill some principles about MAPK signaling and function. Understanding MAPK signaling pathways gives us insight about how cells take decisions in front of stimuli. It is certainly a rewarding activity for a biologist.

## Figures and Tables

**Figure 1 ijms-21-02346-f001:**
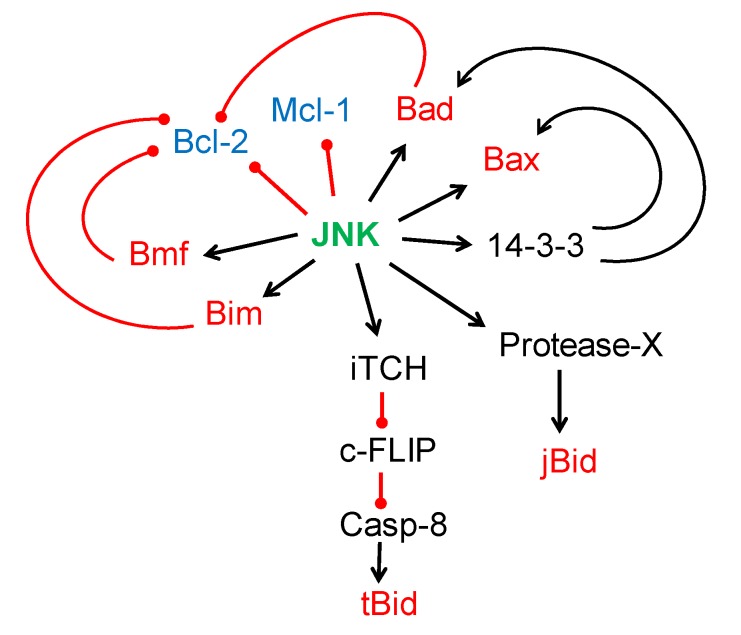
JNK (C-Jun N-terminal kinase) substrates that regulate the mitochondrial apoptotic pathway. JNK can have pro-apoptotic effects through direct phosphorylation of Bcl-2 (B-cell lymphoma 2) family members. The phosphorylation of Bax (Bcl-2-associated X protein) will activate the pro-apoptotic activity of this protein, whereas the phosphorylation of Bcl-2 and Mcl-1 (myeloid cell leukemia 1) will suppress their anti-apoptotic function. Indeed, the direct phosphorylation and activation of Bad (Bcl-2-antagonist of cell death), Bim (Bcl-2-like protein 11), or Bmf (Bcl-2-modifying factor) will inhibit the antiapoptotic effect of Bcl-2. In addition, JNK phosphorylation of 14-3-3 protein induces the release of pro-apoptotic proteins Bax and Bad. JNK phosphorylation of the E3 ubiquitin ligase iTCH (itchy homolog) promotes the degradation of caspase-8 inhibitor c-FLIP (cellular FLICE (FADD-like IL-1β converting enzyme) inhibitory protein), therefore promoting caspase-8 activation and Bid (BH3 interacting-domain death agonist) proteolysis (tBid). JNK, activated by TNFα (tumor necrosis factor alfa), also promotes apoptosis through proteolysis of Bid (jBid) by an unknown protease (X). All these modifications in the Bcl-2 family members induce the release of cytochrome c and/or Smac/DIABLO (second mitochondria-derived activator of caspases/direct inhibitor of apoptosis protein-binding protein with low pI) from the mitochondria and subsequent caspase activation. Symbols used: activation (→), inhibition (―•). Pro-apoptotic members of the Bcl-2 family are represented in red and anti-apoptotic members in blue.

**Figure 2 ijms-21-02346-f002:**
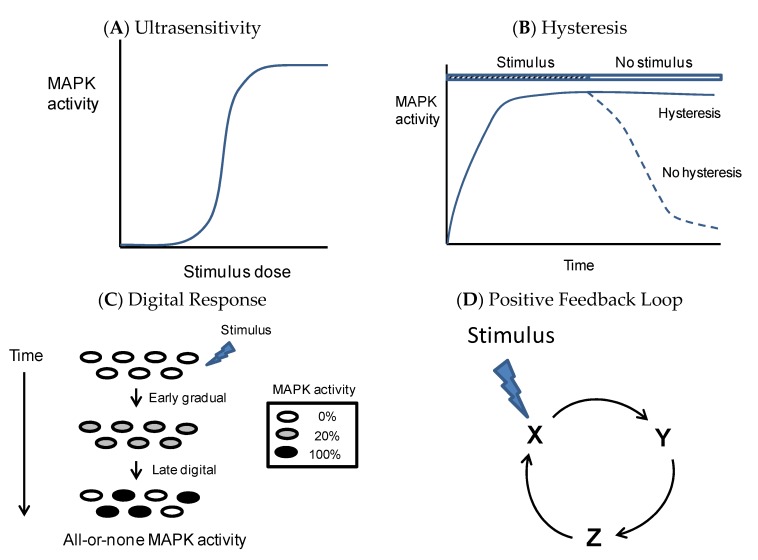
Basic properties of MAPK (mitogen-activated protein kinase) signaling pathways. (**A**) Ultrasensitivity: A small increase in the stimulus produces a very large response in MAPK activity. Graphically, this is illustrated with a sigmoid curve, implying a threshold level for MAPK activation. Several biochemical mechanisms can explain this behavior. (**B**) Hysteresis: MAPKs in front of a certain stimulus, and under certain conditions, present sustained activation even when the stimulus has disappeared but under different conditions or stimulus, could return to basal levels (no hysteresis). (**C**) Digital response: single cells exposed to a certain dose of stimulus will present an initial homogeneous activation of MAPK (20% activity, graded response), which could be converted into an all-or-none response (0% or 100% activity, digital response) at a later time. The generation of digital responses in stress sensors depends on time, threshold levels of activation, ultrasensitivity, and the presence of positive feedback loops. (**D**) Positive feedback-loop: An initial stimulus will activate X, which in turn will activate Y, which will feed back to the input X. These loops, that once engaged would be independent of the initial stimulus, are commonplace in irreversible biological processes.

**Figure 3 ijms-21-02346-f003:**
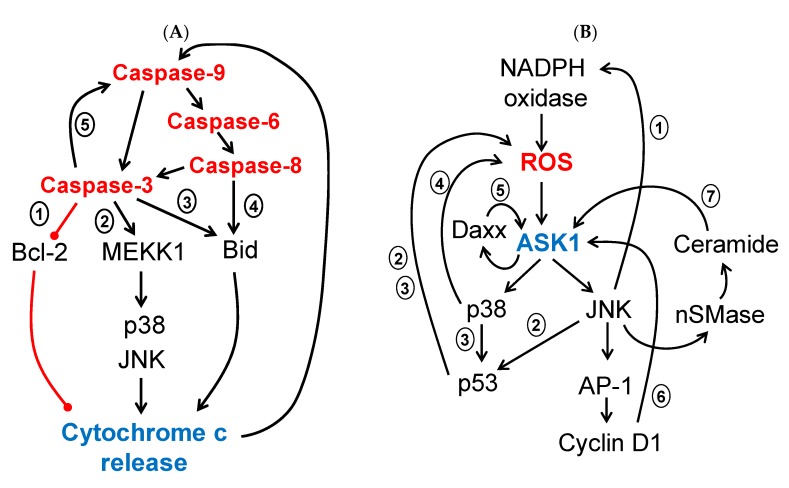
Positive feedback loops in apoptosis. (**A**) Caspases, kinases, and Bcl-2 family members are functionally linked in positive feedback loops to induce cell death. (**1**) Cytochrome c release from the mitochondria, a meeting point in the cell death pathways, induces caspase activation. Caspase-9 and subsequent caspase-3 activation produces Bcl-2 proteolysis, which in turn promotes more cytochrome c release [108]. (**2**) Caspase-3 induces proteolysis and constitutive activation of MEKK1 (MAPK/ERK kinase kinase 1), that promotes cytochrome c release and caspase-9/-3 activation though JNK/p38 activation [119,120,121]. (**3** and **4**) Caspase-3 [110,111] or caspase-8 [109] induce Bid proteolysis (tBid), which in turn promotes cytochrome c release and caspase activation. (**5**) Caspase-3 is activated by caspase-9, which in turn can induce proteolysis and full activation of caspase-9 [112,113]. (**B**) ROS (reactive oxygen species) generation and ASK1 (apoptosis signal-regulating kinase 1) activation are sustained by multiple positive feedback loops. (**1**) NADPH oxidase generates ROS and activates ASK1 and JNK, which in turn activates NADPH oxidase [122,123]. (**2** and **3**) JNK and/or p38 activation, induced by ROS, produces p53 activation, which in turn induces ROS generation [27,124]. (**4**) Sustained activation of p38 increases ROS production, which in turn activates ASK1 and p38 [125,126]. (**5**) ASK1 activation induces the accumulation of Daxx protein, which in turn further activates ASK1 [129]. (**6**) ASK1 induces transcription of cyclin D1 through AP-1 (activator protein 1) activation, which in turn increases ASK1 levels via the Rb-E2F (retinoblastoma-E2F transcription factor) pathway [130]. (**7**) Ceramide activates the ASK1 and JNK signaling pathway, which in turn activates neutral sphingomyelinase (nSMase), thus increasing ceramide production [131,132]. The loops represented in the figure do not necessarily take place in the same cell and at the same time. Some loops are activated by a specific stimulus or in a certain cell type. Symbols used: activation (→), inhibition (―•).

**Figure 4 ijms-21-02346-f004:**
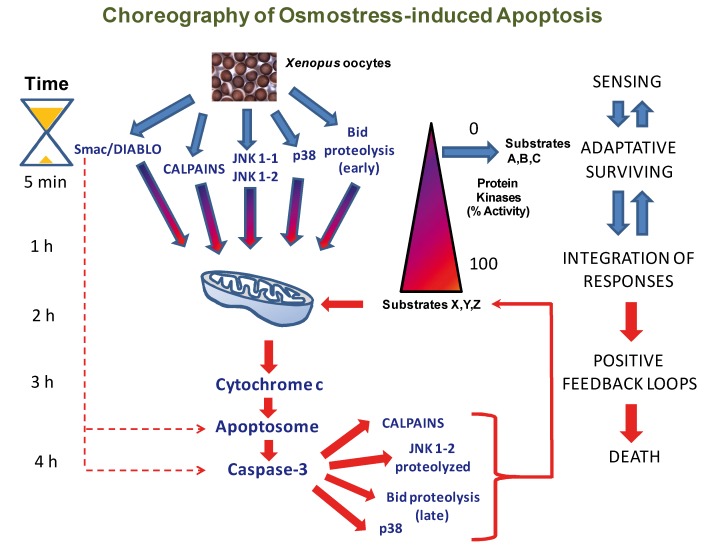
A model for osmostress-induced apoptosis. In *Xenopus* oocytes, hyperosmotic shock induces rapid calpain activation, Smac/DIABLO release from the mitochondria, cleavage of small amounts of Bid by an unknown protease, and MAPK activation (JNK1-1, JNK1-2, and p38). Low amounts of cytochrome c are released from the mitochondria at this early stage (0–1 h), but they are not sufficient to activate caspase-3. Stress protein kinases (JNKs and p38) would act as cellular sensors to evaluate the stressful situation. MAPKs can engage, by early phosphorylation of some substrates (A, B, C), a protective response. Mitochondria integrates the information received by stress sensors at this early stage, where the cell could recover if the stress does not persist or gets weaker. However, sustained and increased activation of MAPKs will lead to late phosphorylation and activation of pro-apoptotic substrates (X, Y, Z), including Bcl-2 family members. A marked release of cytochrome c will promote caspase-3 activation (2 h), which in turn would induce more calpain activation, cleavage of JNK1-2 and Bid, and p38 activation. These events promote additional cytochrome c release and caspase-3 activation in multiple positive feedback loops, resulting in an irreversible apoptotic process (2–4 h). The blue color in the arrows represents early responses to stress in the reversible phase of apoptosis, and the red color represents late responses in the irreversible phase.

**Figure 5 ijms-21-02346-f005:**
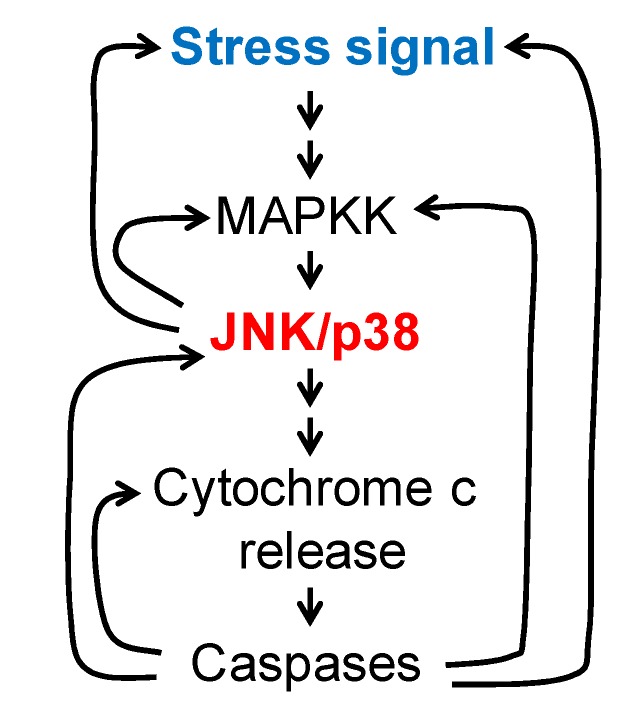
MAPKs in apoptosis: A simplified model. An input, represented by a stress signal (Ca^2+^, ROS, osmotic shock, death receptor, etc.), will activate a MAPK cascade, which in turn feedbacks on the initial input/cascade to achieve higher MAPK activation. After a threshold of MAPK activation is surpassed, additional downstream targets are activated, promoting cytochrome c release and caspase activation. Other positive feedback loops, engaged downstream of MAPKs, will keep sustained and high levels of activation of JNK/p38 and caspases, assuring the irreversibility of the apoptotic process.

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
