# Peer review of "Understanding MAPK Signaling Pathways in Apoptosis"

_ijms, 2020, doi:10.3390/ijms21072346_

Round 1

Reviewer 1 Report

In the paper titled: Understanding MAPK Signaling Pathway in Apoptosis and Meiosis, authors take a difficult task to explain the wiring and feedback loops governing MAPK signaling in normal cells during hyperosmolarity induced stress or during meiosis. The paper is recommended for publication, as it presents unique approach to signaling and to MPAK induced programmed cell death depending on the strength, longevity, as well as on the type of cells and type of stressor with intensity of initial signal. The wiring is illustrated and supported by several Figures with explanation of each Figure presented in the text.

Author Response

Thank you very much for the positive evaluation of our manuscript.

José M. López, Ph. D.

Institut de Neurociències

Departament de Bioquímica i Biologia Molecular

Unitat de Bioquímica. Facultad de Medicina

Universitat Autònoma de Barcelona

  1. Cerdanyola del Vallès

Barcelona. Spain

Phone: (+34) 935814278

Email: josemanuel.lopez@uab.cat

Reviewer 2 Report

In the manuscript titled: “Understanding MAPK Signaling Pathways in 2 Apoptosis and Meiosis” Yue and López aim to review the way MAP Kinase activity is involved in apoptosis and meiosis with an emphasis on Xenopus oocytes. They highlight common mechanisms, riddles and contradictions, and offer models that can explain and generalize these processes. They focus their discussion on: JNK, p38 and ERK1/2, and refrain from addressing the roles of other MAPK’s. In the first part of the review they describe the roles of MAPK on Apoptosis, and in the second they turn to the roles of MAPK during the reductional cell division, meiosis (hence forward, part 1 and part 2 respectively).

Since its discovery, the number of cellular and developmental processes which are activated through the MAPK signaling has been growing. Some of these are being executed simultaneously, and in some cases the MAPK signaling can lead to opposite outcomes. These facts have brought to light a standing question in the field: what is the mechanism which allows the same signaling cascade to reach the desired outcome?  In this respect the choice of apoptosis and meiosis is wise, for both are intertwined, and in both MAPK plays a critical role as a driver and a blocker.

In part 1, the authors first discuss how JNK and p38 drive apoptosis through transcriptional and post transcriptional targets, and then go on to present reports which indicate they work to halt apoptosis. After repeating this processes with ERK (only this time starting with halting) they review the general features of MAPK signaling at the cellular and signaling levels, and using these they offer a possible model that can explain how MAPK can both activate and block apoptosis. This is mainly explained through level and length of the signaling.

After a short introduction to meiosis, they go back to apoptosis and this time focus on Xenopus oocytes.

Generally speaking part 1 is well written, gives ample examples of the authors claims, utilize the figures to illustrate their points and generalizations, and the model they present is reasonable and may serve others who work in the field (see also specific comments). The only major aspect that is missing in this part is reference to Drosophila and C. elegans through which we learned much on the genetic base through which apoptosis is activated. Nevertheless, given the space constrains this is only “nice to have” and this part can certainly be published without it.

Part two is focused on the roles of MAPK in meiosis after the release of Xenopus oocytes from MI arrest. They describe how progesterone start the process which also includes ERK2, and describe the network it is part of.

Part 2, unlike part 1, is extremely short (~ 2 pages) contain unsupported claims, miss most of the roles of MAPK during meiosis, and in fact miss most of meiosis. Indeed, the authors have chosen to discuss meiosis in the context of a system which mostly allows to follow meiosis only after MI, whereas the heart of meiosis lies in prophase I. During prophase I several unique meiotic processes occur: homolog pairing, synapsis, Spo11 mediated double strand breaks formation, and repair through unique players that, unlike all other cellular stages, give rise to interhomolog crossovers. It is therefore not surprising that in many systems prophase I is the longest phase of meiosis. Moreover, the connection between apoptosis and meiosis is mostly present during prophase I. Indeed, in almost all animals tested, specific meiotic prophase I checkpoints exist to send defective meiocytes into apoptosis. As MAPK was shown to affect prophase I progression as well as meiotic apoptosis, it is at that phase that the dilemma the authors present at the beginning comes to full term.

The claims that the authors raise in regard to the central role of MAPK in meiosis, and its opposite function, as well as switchable nature, are poorly supported (and most of the support comes from their own work)). It is this reviewer’s assessment that the discussion of MAPK roles after MI as presented in the manuscript contribute little to the review, its main points, and the model suggested. Limiting the review to post Mi stages may be misleading, and therefore this reviewer recommends accepting the manuscripts only under two scenarios:

  1. Omitting part two and only discuss the roles of MAPK in apoptosis. This part can very well stand by itself and serve the readers well.
  2. Completely rewrite part 2, choose another system (e.g. mammalian spermatogenesis, D. melanogaster or C. elegans gonads, plenty of reviews discus MAPK roles in these), review the roles and signaling pathway there, and the way this feed into apoptosis and their model.

Point by point comments:

  1. Line 70 and others. It is generally accepted when citing review papers to mention this fact (“reviewed in….”).
  2. Lines 110-112. This sentence is unclear.
  3. Figure 1 (and other figures as well). It will be better to explain the color code in the legend.
  4. Lines 270-271. I think the authors have mixed this up, as the result suggest otherwise. See Fig 4 in Persons, et al.
  5. Line 430 and onwards. I find the authors decision to describe positive feedback loops as “strange loops” surprising. Contrary to their claim, positive feedback loops are known in many fields of biology, and have been discussed in length (see for example Ingolia and Murray, 2007). I would stick with the accepted term “positive feedback” to avoid confusion.
  6. Lines 438-451. The very fact that during apoptosis several caspases auto-activate or cross activate is another example of positive feedback.
  7. Figure 3 and elsewhere. A list of abbreviations would be helpful as the authors use them without introduction.
  8. Section 7. The title is: “Xenopus Oocytes as a Cell Model to Understand Apoptosis and Meiosis”. This seems to be out of place as the authors go back to apoptosis in section 8. Instead this reviewer recommends discussing here Xenopus oocyte as a scientific system in general.
  9. Lines 528-529. This statement is not even close to the history of meiosis studies. By far the system which taught us most about meiosis is S. cerevisiae. As the authors stated the Xenopus oocytes when used are arrested at MI, and thus we cannot learn about the longest and most unique stage – prophase I. I recommend that the authors would look at the titles of the talks in the GRC and EMBO meiosis meetings of the last two decades to learn about the accepted models in the field.
  10. Lines 652-653 the factors that lead S. cerevisiae to meiosis under natural conditions are unclear.
  11. Line 659. C. elegans should be in italics
  12. Lines 665-666. It is unclear how the statements made above led to this conclusion.

Author Response

Barcelona, 10 March 2020

Cover Letter                                                                        

Manuscript ID: ijms-725828

Type of manuscript: Review

Title: Understanding MAPK Signaling Pathways in Apoptosis and Meiosis

Authors: Jicheng Yue, José M. López

Dear Dr. Liu:

   We have revised our manuscript according to the reviewers’ comments. Reviewer 1 agreed with the publication of the manuscript and did not raise any criticism. Reviewer 2 considered that part 1 (MAPK signaling in apoptosis), which occupies most of the manuscript, was well written, could serve others who work in the field, and can be published. However, part 2 (MAPK signaling in meiosis) is short, compared with part 1, and only focus in a specific window of meiosis (from prophase G2/M arrest to MI) using Xenopus oocytes as a cell model. As the reviewer 2 points, meiosis is a complex process and it would deserve a deeper review by itself, analyzing the role of MAPKs during the different phases of meiosis, with more emphasis in prophase I (where apoptosis seems to be relevant), and considering different animal models. Therefore, we have followed the advice of Reviewer 2 and present in our manuscript only part 1, which contains enough information for a review paper. I also think that focusing on apoptosis makes the manuscript more clear for the readers. Thus, we have deleted sections 9, 10, 11, 12, and Figures 5 and 6B from the previous manuscript. We have changed the title, the abstract, and modified some paragraphs in different sections. Figures 3A, 4, and 5 (previous Figure 6A) present only minor changes. A few new references have been included. We only mention, at the end of the manuscript, that the model presented in Figure 5 may apply to other biological processes. Thus, sustained ERK and MPF activation, due to the presence of multiple positive feedback loops, induce meiotic progression in Xenopus oocytes. I think these changes have improved the quality and clarity of the manuscript.

   I uploaded our new version of the manuscript and a copy of the original manuscript marked with the changes to facilitate the review process. I hope the revised version of our manuscript is now acceptable for publication in International Journal of Molecular Sciences.

   Next, following the comments of the Reviewer 2, we will answer point-by-point the issues raised in the previous letter:

  1. Line 70 and others. It is generally accepted when citing review papers to mention this fact (“reviewed in….”).

     Reference 12 is now cited as “reviewed in 12” (line 68 final version or line 72 track changes). Other reviews that appear in our manuscript have been modified to include the expression “reviewed in…”. Thus, references 8, 9, 10, 67, 68, 103 appear in lines 43, 44, 51, 260, 261, 431 in the final version, or in lines 46, 47, 54, 270, 271, 442 in track changes).

  1. Lines 110-112. This sentence is unclear.

The paragraph “Different to JNK, p38 MAPK promotes cell death through phosphorylation of p53 at Ser46 instead of Thr81 in HIV-1 envelope glycoprotein complex (Env) induced syncytia [25], which may imply a distinct dimerization status” has been changed to “However, in HIV-1 envelope glycoprotein complex (Env)-induced apoptosis p38 MAPK promotes cell death through phosphorylation of p53 at Ser46 instead of Thr81 [25], which may imply a distinct dimerization status of p53” to improve clarity. Lines 108-110 final version or lines 112-115 in track changes.

  1. Figure 1 (and other figures as well). It will be better to explain the color code in the legend.

     In Figure 1 we have added the following sentence: “Pro-apoptotic members of Bcl-2 family are represented in red and anti-apoptotic members in blue” to explain color code for the Bcl-2 family members (lines 201-202 final version or lines 211-212 in track changes). In Figure 4 we included the sentence: “Blue color in arrows represents early responses to stress in the reversible phase of apoptosis, and red color represents late responses in the irreversible phase” to explain color code for early/reversible and late/irreversible events during apoptosis (lines 653-654 final version or lines 689-690 in track changes).

  1. Lines 270-271. I think the authors have mixed this up, as the result suggest otherwise. See Fig 4 in Persons, et al.

     The paragraph has been corrected. DNA damage induced by cisplatin activates ERK1/2 and increases p53 levels [73]. The authors also found that ERK1/2 interacts with p53 and induces the phosphorylation of p53 at Ser15. Therefore, upregulation of p53 by ERK signaling pathway may be one of the mechanisms of DNA damage-induced apoptosis. Lines 270-274 final version or lines 280-285 in track changes.

  1. Line 430 and onwards. I find the authors decision to describe positive feedback loops as “strange loops” surprising. Contrary to their claim, positive feedback loops are known in many fields of biology, and have been discussed in length (see for example Ingolia and Murray, 2007). I would stick with the accepted term “positive feedback” to avoid confusion.

     The term “strange loop” of signaling has been eliminated along the manuscript. The reviewer is right pointing that positive feedback loops are known in many fields of biology. Accordingly, we have rewritten this section, including new references. Lines 428-440 final version or lines 439-455 in track changes.

  1. Lines 438-451. The very fact that during apoptosis several caspases auto-activate or cross activate is another example of positive feedback.

     Auto-activation of apical caspases (caspase-9 or caspase-8) by aggregation in multiprotein complexes is not considered a positive feedback loop. However, it is true that more cross-activation between caspases is possible. We have included activation of caspase-9 by caspase-3, although there is some discussion in the field if this must be considered a derepression mechanism or a positive feedback loop (see references 112-114, included now in the manuscript). We have included also caspase-3 activation by caspase-8 (new reference 115) (Figure 3A). A recent study, which we have included in the new version, emphasizes the important role of effector caspases engaging positive feedback loops for efficient cell death (new reference 116). Lines 452-464 final version or lines 467-479 in track changes.

  1. Figure 3 and elsewhere. A list of abbreviations would be helpful as the authors use them without introduction.

     We have revised the manuscript and defined the abbreviations in parentheses the first time they appear in the text, as required by the guidelines of IJMS. Here is the list of abbreviations that we have included in the revised manuscript and that were not present in the previous version:

Line 13 (line 13 track). MAPK: mitogen-activated protein kinase

Line 75 (line 79 track).  Bcl-2: B-cell lymphoma 2

Line 115 (line 120 track).  Wip1: wild-type p53-induced phosphatase 1

Line 119 (line 124 track). AMPK: 5' adenosine monophosphate-activated protein kinase

Line 123 (line 128 track). FoxO1: forkhead box protein O1

Line 133 (line 138 track). Bid: BH3 interacting-domain death agonist

Line 138 (line 143 track). iTCH: itchy homolog

Line 139 (line 144 track). c-FLIP: cellular FLICE (FADD-like IL-1β-converting enzyme)-inhibitory protein

Line 143 (line 148 track). Smac/DIABLO: second mitochondria-derived activator of caspases/direct inhibitor of apoptosis protein binding protein with low PI

Line 148 (line 153 track). RIPK: receptor-interacting serine/threonine-protein kinase

Line 157 (line 162 track). Bax: Bcl-2-associated X protein

Line 158 (line 163 track).  Bad: Bcl-2-antagonist of cell death

Line 162 (line 167 track). Bim: Bcl-2-like protein 11

Line 164 (line 169 track). Bmf: Bcl-2-modifying factor

Line 169 (line 174 track). Mcl-1: myeloid cell leukemia 1

Line 171 (line 176 track). Bcl-xL: B-cell lymphoma-extra large

Line 286 (line 297 track). Bak: Bcl-2 homologous antagonist killer

Line 364 (line 375 track). PDGF: Platelet-derived growth factor

Line 371 (line 382 track). Hog1: high osmolarity glicerol 1

Line 466 (line 481 track). MEKK1: MAPK/ERK kinase kinase 1

Line 525 (line 544 track). ASK1: apoptosis signal-regulating kinase 1

Line 528 (line 547 track). TNFR1: tumor necrosis factor receptor 1

Line 532 (line 551 track). Daxx: death-associated protein 6

Comments 8-12 refer to part 2 (meiosis) in our previous manuscript, but this part has been deleted in the new version, and therefore we have not addressed these points.

Sincerely

José M. López, Ph. D.

Institut de Neurociències

Departament de Bioquímica i Biologia Molecular

Unitat de Bioquímica. Facultad de Medicina

Universitat Autònoma de Barcelona

  1. Cerdanyola del Vallès

Barcelona. Spain

Phone: (+34) 935814278

Email: josemanuel.lopez@uab.cat

Round 2

Reviewer 2 Report

In the revised manuscript titled: “Understanding MAPK Signaling Pathways in Apoptosis” Yue and López implemented the vast majority of the comments regarding the original manuscript. The result is a coherent, focused, and insightful work. Limiting the scope to apoptosis results with a body of work that would be of service to a wide audience interested in the place of MAPK in this programmed cell death program. I would like to congratulate the authors for this fine work and have no further comments.